# Synergistic Action of Membrane-Bound and Water-Soluble Antioxidants in Neuroprotection

**DOI:** 10.3390/molecules26175385

**Published:** 2021-09-04

**Authors:** Stephanie K. Polutchko, Gabrielle N. E. Glime, Barbara Demmig-Adams

**Affiliations:** Department of Ecology and Evolutionary Biology, University of Colorado, Boulder, CO 80309-0334, USA; Stephanie.Polutchko@Colorado.EDU (S.K.P.); Gabrielle.Glime@colorado.edu (G.N.E.G.)

**Keywords:** carotenoid, human health, human nutrition, lutein, ROS, zeaxanthin

## Abstract

Prevention of neurodegeneration during aging, and support of optimal brain function throughout the lifespan, requires protection of membrane structure and function. We review the synergistic action of different classes of dietary micronutrients, as well as further synergistic contributions from exercise and stress reduction, in supporting membrane structure and function. We address membrane-associated inflammation involving reactive oxygen species (ROS) that produce immune regulators from polyunsaturated fatty acids (PUFAs) of membrane phospholipids. The potential of dietary micronutrients to maintain membrane fluidity and prevent chronic inflammation is examined with a focus on synergistically acting membrane-soluble components (zeaxanthin, lutein, vitamin E, and omega-3 PUFAs) and water-soluble components (vitamin C and various phenolics). These different classes of micronutrients apparently operate in a series of intertwined oxidation-reduction cycles to protect membrane function and prevent chronic inflammation. At this time, it appears that combinations of a balanced diet with regular moderate exercise and stress-reduction practices are particularly beneficial. Effective whole-food-based diets include the Mediterranean and the MIND diet (Mediterranean-DASH Intervention for Neurodegenerative Delay diet, where DASH stands for Dietary Approaches to Stop Hypertension).

## 1. Introduction and Overview

Humans need micronutrients to assist with, and protect, vital functions. Essential micronutrients—which cannot be synthesized in the human body and must be consumed with the diet—include not only vitamins but also a host of other compounds of plant origin termed phytochemicals (plant chemicals). Phytochemicals can be classified by solubility into hydrophobic, or membrane-soluble, compounds and hydrophilic compounds soluble in aqueous cell environments. Plants produce hundreds of largely water-soluble compounds [1], as well as a handful of water-insoluble, membrane-localized antioxidants including vitamin E (tocopherols) and carotenoids such as zeaxanthin and lutein [2,3,4]. Carotenoids are long-chain terpenoids produced by plants, and also by some bacteria and fungi. Zeaxanthin and lutein are oxygen-containing carotenoids, or xanthophylls, with unique roles in human health [5,6].

Membranes are the site of neuronal function and are thus abundant in the mammalian brain. For both neuronal and other functions, membranes need to have a specific fluidity. Membrane fluidity is influenced by the type of fatty acids in the phospholipids that make up membranes and can be further modulated by metabolites that are dissolved within these membranes. A well-known example of such a metabolite is cholesterol that acts as a stabilizer of membrane fluidity. The dietary xanthophylls, zeaxanthin and lutein, also act as membrane stabilizers in human membranes (for a recent review, see [5]; see also further detail and Figure 1 below). Several carotenoids and vitamin E are present in the human brain, with xanthophylls accounting for up to three-quarters of total carotenoids in the brain [7]. These xanthophylls are mainly zeaxanthin and lutein, with a high ratio of zeaxanthin to lutein [7]. The frontal cortex, the site of executive function and particularly vulnerable to, e.g., Alzheimer′s disease, has a particularly high concentration of these xanthophylls [7].

Furthermore, membranes are a key site in the initiation of an immune response. When a series of events produce oxidants, polyunsaturated fatty acids (PUFAs) of membrane phospholipids become oxidized to hormone-like gene regulators that either *trigger* an immune response (generally via derivatives of omega-6 PUFAs) or *terminate* the immune response (generally via derivatives of omega-3 PUFAs) [8]. Conversely, an active immune response produces reactive oxygen species (ROS) to help eliminate pathogens and to serve as signals that prompt further activation of the immune system [9]. A variety of environmental and lifestyle factors can stimulate ROS production even further or can contribute to a deficiency in the antioxidants that keep ROS in check. Such an imbalance between oxidants and antioxidants contributes to continuous activation of the immune response, or chronic inflammation (see sections dedicated to these topics below). Chronic inflammation in the brain, or neuroinflammation, is a risk factor for several diseases and disorders as well as for a state of low mental function in otherwise healthy individuals. The brain contains not only many polyunsaturated fatty acids that are highly susceptible to oxidation, but also exhibits high levels of oxygen. In addition, some causes of neuroinflammation reside in sites far from the brain. For example, an impaired gut barrier allows bacterial toxin, lipopolysaccharide (LPS), to cross from the intestine into the bloodstream and trigger LPS-induced neuroinflammation (see details on this topic below).

Several classes of phytochemicals must work in tandem with each other to prevent excessive oxidation of the fatty acids of membrane phospholipids (see details on this topic below). A direct interaction of membrane-soluble and water-soluble antioxidants was demonstrated by in vitro studies with isolated membrane vesicles. More research is needed to clarify if, and where, in the human body dietary phytochemicals co-occur and interact directly in the same sites in vivo. Vitamin E, zeaxanthin, and lutein are readily taken up from the gut and cross the blood–brain barrier (e.g., [7]). It was shown that supplementation with lutein lowered the level of LPS-induced neuroinflammation [10]. In addition, vitamin E is not only found in the brain but also promotes the movement of dietary polyphenols across the blood-brain barrier (as shown for the flavonol quercetin as well as for rutin that consists of quercetin and the disaccharide rutinose [11]). However, even phytochemicals that do not cross the blood–brain barrier can contribute to preventing LPS-associated neuroinflammation [12] (for details, see below).

## 2. From the Structure of Phytochemicals to Their Functions

Figure 1 depicts how zeaxanthin and lutein incorporate themselves into biological membranes with their hydrophobic middle portions (in orange) mixing with the fatty acid tails of membrane phospholipid (in yellow) and the hydrophilic end structures (in dark blue) aligning with the hydrophilic portions of membrane phospholipids (in light blue). This scheme illustrates the critical importance of the oxygen-containing portions of these two xanthophylls in determining their localization in the membrane, and ability to stabilize membranes, which differs from the localization and function of oxygen-free carotenes that float freely in the center of the membrane in their entirety (see [5]). In addition to serving as membrane stabilizers, both zeaxanthin and lutein are potent antioxidants due to their long system of conjugated C=C double bonds. Zeaxanthin is the more potent antioxidant of the two by virtue of its slightly longer system of conjugated C=C double bonds (all 11 C=C double bonds are in conjugation in zeaxanthin, whereas only 10 of the 11 are in conjugation in lutein). Zeaxanthin has also been proposed to be the more potent membrane stabilizer of the two xanthophylls (Figure 2; see [5]).

Like the xanthophylls, vitamin E also has both hydrophobic and hydrophilic portions (Figure 2) and is incorporated into the phospholipid bilayer accordingly. However, unlike the two xanthophylls, vitamin E does not span the whole width of the membrane and thus does not have the same membrane-stabilizing effect (Figure 1).

Zeaxanthin and vitamin E are both capable of reducing, and thereby detoxifying, oxidants (such as ROS) directly as well as reducing lipid peroxyl radicals, which slows the chain reaction of lipid oxidation induced by ROS [13]. While zeaxanthin alone is rapidly consumed in this process, a combination of zeaxanthin and vitamin E provides prolonged protection [13] based on recycling (re-reduction) of zeaxanthin radical by vitamin E via donation of an electron [14] (Figure 3a). In addition, either zeaxanthin or vitamin E radicals can be recycled via re-reduction by water-soluble antioxidants at the membrane interface with aqueous cellular spaces [14,15] (Figure 3b,c). Figure 3 also depicts this recycling of oxidized vitamin E or zeaxanthin radical by water-soluble antioxidants at the membrane-cytosol interface (Figure 3a–c). For example, the water-soluble dietary vitamin C (ascorbic acid) can re-reduce vitamin E radical (α-chromanoxyl radical [16,17]) and can also reduce zeaxanthin radical [13,14,18] (Figure 3). In fact, it is critical that vitamin E and zeaxanthin radicals be recycled (by re-reduction) because they themselves are potentially harmful oxidants [18,19]. The importance of this recycling process is illustrated by the fact that the antioxidant capacity of various vitamin E constituents is determined by how efficiently they can be recycled [19]. Future research is needed to ascertain whether one of the possible combinations of cycles shown in Figure 3 for how zeaxanthin and/or vitamin E radicals can be recycled (Figure 3a–c) may be more prominent than the others. Each of these cycles ends in reduction of the radical form of a membrane-soluble antioxidant by a water-soluble antioxidant at the membrane–cytosol interface. In addition to dietary water-insoluble antioxidants, endogenous water-soluble antioxidant metabolites and water-soluble antioxidant enzymes can both contribute to this important last step in the re-reduction cascade. An example for involvement of both metabolites and enzymes is the recycling/re-reduction of tocopherol radical by the metabolite glutathione in cooperation with (as a cofactor of) antioxidant enzymes [20]. Finally, the oxidized forms of water-soluble antioxidant metabolites can be recycled (re-reduced) by electron donors of which cells produces a near-infinite supply, such as the electron-and-hydrogen donor nicotinamide adenine dinucleotide (NADH) that can re-reduce oxidized vitamin C [21]. The multiple intertwined cycles of reduction and oxidation can be viewed as a way to extend the lifetime of dietary antioxidants by ultimately recycling them via endogenous electron donors available in ample supply. Since NADH is not membrane soluble, its utilization requires a sequence of reactions suitable to tie NADH to dietary antioxidant metabolites that are lipid-soluble and control the part membrane-localized component of the inflammation process.

More research is needed to clarify how these interactions occur in vivo with respect to the relative contribution of the different cycles shown in Figure 3 and to the relative contribution of non-enzymatic versus enzymatic mechanisms. In addition to interacting non-enzymatically with either lipids or their oxidized radicals, both oxidants and antioxidants can directly modulate the activity of enzymes that oxidize membrane lipids to hormone-like messengers. An example is the enzyme lipoxygenase that is activated by ROS and inhibited by tocopherol or phenolics like red-wine resveratrol [22,23]. Through this control of enzymatic and non-enzymatic production of messengers, dietary antioxidants modulate key signal transduction pathways and gene expression patterns [24].

Water-soluble dietary phytochemicals with antioxidant capacity include vitamin C and the large, diverse group of phenolics (for a review, see [1]). Phenolics contain one or more phenolic ring structure(s) with a system of conjugated carbon–carbon double bonds (C=C) that supports loss of an electron without losing stability, which is a hallmark feature of potent antioxidants [25]. Many phenolics are water-soluble due to their hydrophilic side groups. A major subgroup of phenolics are the flavonoids that comprise multiple subgroups, including flavanols (catechins) and anthocyanidins (the phenolic portion of the sugar-containing anthocyanins [1,26]).

Evidence for direct interaction between membrane-bound vitamin E or carotenoids on the one hand and water-soluble antioxidants on the other hand mainly comes from in vitro studies that showed direct interaction between membrane-soluble and water-soluble phytochemicals in preventing oxidation of membrane phospholipids. For example, the ability to re-reduce vitamin E and zeaxanthin radicals was documented for ascorbic acid (vitamin C; see above) as well as phenolics such as anthocyanidins (that can interact with zeaxanthin [15]), catechins (that can interact with vitamin E [27]), or rutin (that can interact with lutein [28]).

Future research should elucidate how the many phenolic phytochemicals of a plant-rich diet interact synergistically with membrane-bound phytochemicals in vivo, as there are several possibilities of how this may occur. While many polyphenols are absorbed poorly beyond the gut in their original forms [25], some polyphenols may move beyond the intestinal tract and across the blood–brain barrier [29,30]. Certain polyphenols are broken down by gut bacteria into metabolites that move out of the gut and exert a variety of anti-inflammatory and antioxidative effects, as has been suggested for metabolites of anthocyanins and other flavonoids [31,32,33,34]. Both green tea extract [35] and proanthocyanidins from grape seeds [36] decreased the levels of lipid hydroperoxides in blood plasma, and both red wine polyphenols [37] and cocoa flavanols [38] reduced the circulating levels of gene regulators produced by PUFA oxidation.

Additional indirect interaction is possible between membrane-soluble antioxidants (located in membranes throughout the body) and water-soluble polyphenols that remain in the intestinal tract. For phenolic phytochemicals that do not move beyond the intestinal tract, an alternative mechanism exists that allows them to exert synergistic effects with membrane-bound phytochemicals throughout the body (Figure 3d). This alternative mechanism involves the gut–brain axis. Dietary phenolics and other phytochemicals have a key role in the intestinal tract in maintaining gut microbiome eubiosis (a beneficial, diverse microbiome composition) and gut barrier integrity [39,40]. Evidence is mounting that a functional gut barrier prevents translocation of LPS into the bloodstream and resulting LPS-induced neuroinflammation in neurodegenerative diseases, such as dementia and Alzheimer′s disease [41]; see also [42,43,44]. Protection of gut barrier integrity by intestinal phytochemicals may ameliorate neuroinflammation, and thereby act synergistically with membrane-bound phytochemicals in the brain [45]. In addition, translocation of metabolites of dietary phenolics out of the gut appears to play a role and requires further study. A recent review [45] specifically addressed the ability of small (low-molecular-weight) metabolites of polyphenols to attenuate neuroinflammation.

The available evidence is consistent with an ability of zeaxanthin and/or lutein to attenuate neuroinflammation [10,46]. Irrespective of whether various different classes of phytochemicals act synergistically within the brain, throughout the body, or in separate sites, phytochemicals should thus be studied, and consumed, in combination with each other.

## 3. Epidemiological and Clinical Evidence

While there is substantial evidence for inverse correlations between zeaxanthin and/or lutein levels in blood or tissues and the risk for various diseases, the levels of these two xanthophylls often correlate closely with those of other carotenoids, including provitamin A (β-carotene) and lycopene (see, e.g., [47]). Carotenoids with pro-vitamin A activity act as immune-modulatory gene regulators through the known effect of vitamin A compounds [48]. To establish causality in disease-risk reduction for the case of zeaxanthin and/or lutein, double-blind, randomized clinical trials are thus needed [49].

There is a growing number of supplementation studies with zeaxanthin and/or lutein. Supplementation with zeaxanthin and lutein enhanced cognitive performance in both young and older adults (e.g., [50,51,52,53]; for a recent overview, see also [5]). In a study on healthy young adults, supplementation with zeaxanthin as well as a mixed formulation containing zeaxanthin, lutein, and mixed *n*-3 fatty acids specifically and significantly increased visual processing speed [50]. While zeaxanthin and lutein levels were each independently associated with enhanced cognition, memory, and executive function, only zeaxanthin, but not lutein, was also associated with greater processing speed in older adults [54]. Randomized, double-blind, placebo-controlled clinical trials have also demonstrated memory-enhancing effects of daily xanthophyll supplementation in healthy middle-aged adults [55] and older adults [51]. Likewise, a recent meta-analysis of clinical trials concluded that carotenoid-based interventions, and especially those using zeaxanthin and/or lutein, improved cognitive performance across the lifespan [56,57].

Given the interaction among several membrane components (i.e., PUFAs, vitamin E, and zeaxanthin/lutein) as well as the need for interaction between membrane-bound and water-soluble antioxidants, clinical trials are needed that examine the efficacy of combinations of multi-component supplements. A supplementation study on a combination of lutein and the omega-3 PUFA DHA (docosahexaenoic acid) showed that this combination synergistically enhanced memory as well as the rate and efficiency of learning [56,58]. Positive outcomes were also demonstrated in patients with Alzheimer′s disease who consumed a combination of xanthophyll carotenoids and omega-3 PUFAs (fish oil [59]). There is also evidence for synergistic action between vitamin E and carotenoids in supporting cognitive performance in middle-aged adults [60]. Supplementation with a combination of omega-3 PUFAs and vitamins C and E also prevented postoperative atrial fibrillation (as a condition that depends on neuronal activity), while omega-3 PUFA alone did not [61,62]. More clinical trials are needed to further address the synergy among various dietary micronutrients. It is possible that best outcomes will require consumption of a mix of a large number of diverse and synergistically acting factors, as provided by a whole-food diet. Diets that include omega-3 PUFAs and multiple antioxidants, such as the Mediterranean diet and the MIND diet (Mediterranean-DASH Intervention for Neurodegenerative Delay, where DASH stands for Dietary Approaches to Stop Hypertension), have been shown to lead to improved clinical outcomes for Alzheimer’s patients [63].

## 4. Physical Activity and Stress Reduction Support Diet in Maintaining Oxidant-Antioxidant Balance

In addition to synergy among diverse dietary components, synergistic interactions take place between diet and other lifestyle factors, such as physical activity and psychological stress. Figure 4 summarizes key interactions of diet with physical activity and stress level in modulating ROS level as well as the resulting formation of immune-regulatory derivatives of membrane phospholipids. While diet must provide essential PUFAs and antioxidant metabolites (see below), physical activity is necessary to trigger synthesis of endogenous antioxidant enzymes, and stress reduction is needed to lower ROS production as a result of chronic psychological stress. Chronic inflammation, therefore, responds to the combination of all of these lifestyle factors [64].

### 4.1. Diet

Vitamins, carotenoids, and PUFAs are all essential micronutrients that cannot be synthesized by humans and must be consumed with the diet. Furthermore, dietary antioxidant metabolites, such as zeaxanthin, lutein, vitamin E, vitamin C, and phenolics, are required at just the right levels to prevent excess accumulation of ROS without interfering with the beneficial roles of ROS in small amounts. Small amounts of ROS are instrumental in orchestrating the immune response, including temporary formation of inflammation-*promoting* gene regulators during an acute infection, as well as formation of inflammation-*terminating* gene regulators after an acute infection is brought under control and to prevent detrimental chronic inflammation (Figure 4). At the bottom of Figure 4, the blue arrow from dietary antioxidants to suppression of inflammation-promoting, PUFA-6-based regulators addresses this beneficial effect of antioxidants. However, the same antioxidants can also negatively impact the immune response by suppressing formation of inflammation-terminating PUFA-3-based regulators (Figure 4, red arrow from antioxidants at the bottom). Such negative impacts are particularly likely when an excess of antioxidants is consumed. Excess antioxidant levels can also suppress ROS formation to the point that antioxidant enzyme synthesis is no longer induced (not shown in Figure 4). In particular, excess antioxidant intake from high-dose supplements has been shown to suppress synthesis of endogenous antioxidant enzymes [65]. A delicate balance (cellular redox homeostasis) between oxidants (such as ROS) and antioxidants is thus required for optimal functioning of metabolism, and excess antioxidant intake clearly has the potential to be detrimental [66].

A balanced intake of antioxidant metabolites also needs to be paired with a balanced intake of omega-6 and omega-3 fatty acids as the precursors of immune regulators (Figure 4). These two types of fatty acids are needed in a ratio that balances production of inflammation-triggering and inflammation-terminating, PUFA-based gene regulators (Figure 4). Over the past century, the ratio in which omega-6 and omega-3 are consumed in industrialized countries has “increased from approximately 4:1 to 20:1” [8]. Figure 4 illustrates how an excessively high ratio of omega-6 to omega-3 fatty acids in the diet interacts synergistically with an imbalance between ROS production and antioxidant capacity in promoting chronic inflammation. The series of intertwined oxidation–reduction cycles described above indicates that concomitant consumption of water-insoluble and water-soluble antioxidants is important to extend the lifetime of membrane-soluble antioxidants. Moreover, the diet must also provide the mineral cofactors (zinc, selenium, copper, and manganese) required by most endogenous antioxidant enzymes. Until the effect of combinations of dietary micronutrients, and the balance in which they are needed, are better understood, “antioxidants are best acquired through whole food consumption [rather than] dietary supplements” [1]. This conclusion probably applies particularly to long-term diet management for disease prevention. We do not wish to exclude that management of existing disease can benefit from evidence-based, expert-monitored supplementation efforts. An active area of research on the benefits of supplementation, with a focus on zeaxanthin, is the prevention of eye disease, such as age-related macular degeneration. Combinations of zeaxanthin and/or lutein with synergistically acting lipid-soluble vitamin E, water-soluble vitamin C, and/or omega-3 polyunsaturated fatty acids (e.g., DHA) were particularly effective [67,68,69]. The mechanisms of action of zeaxanthin and lutein in this context have been documented and reviewed extensively [70,71,72,73,74] (see also [3,5]).

On the other hand, the Mediterranean diet is an example of an anti-inflammatory diet (for a review of the connections and mechanisms listed below, see [75]). This diet is low in saturated fat (that stimulates storage of fat around the waist, which is a source of inflammation-triggering hormones) and high in monounsaturated fats and anti-inflammatory n-3 PUFAs. Additionally, the Mediterranean diet is low in free sugar and quick-burning starches (with a high glycemic index, which also stimulates fat storage around the waist) and high in vegetables (with a low glycemic index), fresh fruits, as well as herbs and spices (powerhouses of phytochemicals). Clinical trials have linked the Mediterranean diet to a reduced risk of depression [76], Alzheimer’s disease [77,78], and other chronic diseases like cardiovascular disease [79]. The mechanisms underlying this effect include amelioration of chronic inflammation as evidenced by lower levels of inflammatory markers [80,81]. Furthermore, the MIND diet was developed specifically to support brain health during the aging process through a combination of the Mediterranean diet and the DASH diet [82,83]. Both the Mediterranean diet and the MIND diet are high in plant-based foods and low in animal/saturated fats. However, the MIND diet focuses comparatively more on consumption of green leafy vegetables (high in multiple phytochemicals) [82] as well as berries with their high concentrations of not only polyphenols that support brain [84] and gut health [85], but also of prebiotic oligosaccharides that support gut microbial diversity [86]. While the Mediterranean and DASH diets can both lower the risk of cognitive decline, adherence to the MIND diet has led to the strongest associations between diet and a reduced risk of Alzheimer’s disease and cognitive decline observed to date [87].

### 4.2. Exercise and Chronic Inflammation

As stated above, synthesis of endogenous antioxidant enzymes is triggered by ROS formed in working muscles and suppressed by high-dose antioxidant supplements (Figure 4; for a review, see [65]). Conversely, physical inactivity fails to produce these essential ROS-based gene regulators needed to trigger the synthesis of antioxidant enzymes. Regular moderate exercise can thus produce a balanced ratio of ROS and antioxidants enzymes, and counteract chronic inflammation, whereas excessive exercise can actually trigger chronic inflammation—presumably by producing ROS at a level too high to be matched by antioxidant enzyme synthesis [88,89]. This sets off a cascade of detrimental events. In the context of brain function, exercise can either trigger or prevent neuroinflammation by effects on the gut–brain axis that depend on exercise intensity, frequency, and duration [90]. Regular moderate exercise supports gut barrier integrity and beneficial gut microbes as well as prevents translocation of LPS into the bloodstream [91,92]. In contrast, excessive exercise compromises the gut barrier and causes gut microbiome dysbiosis [91,92]. More research is needed to define the threshold for excessive exercise for different types of activities and also in the context of individual differences in genetic background, age, and fitness level.

### 4.3. Chronic Psychological Stress

While acute psychological stress, and the associated production of stress hormones, temporarily suppresses the immune system, chronic psychological stress eventually leads to fatigue of the stress response system (the hypothalamus–pituitary–adrenal, or HPA, axis) and results in continuous low-grade activation of the immune system, i.e., chronic inflammation [93]. This fatigue of the stress response system under the influence of chronic psychological stress can involve a lowered mass of stress-hormone-producing glands [94] and unresponsiveness (insensitivity/resistance) of stress-hormone receptors (such as the glucocorticosteroid receptor [95]) to the stress signal. Such disruption of the stress response under chronic psychological stress is somewhat reminiscent of the development of insulin insensitivity, and eventual failure to produce insulin, as a result of chronic high blood glucose. Various stress and sleep management therapies can lower the level of chronic inflammation. Meta-analyses of multiple studies on the effect of yoga [96] (see also [97]) or mindfulness practice [98,99] concluded that these interventions led to a lowering of inflammation biomarkers across multiple human populations studied. Similar effects were reported for cognitive behavioral therapy [100].

### 4.4. Additive or Synergistic Effects

Since diet, physical activity level, and psychological stress level interact to modulate ROS levels and the production of immune-system regulators (Figure 4), a combination of modest changes in each area may be effective in reducing chronic inflammation. The benefits on brain function provided by a combination of several lifestyle factors, including physical activity, cognitive engagement (that promotes stress resilience), and diet, have been reviewed previously [101]. Another recent review of the synergistic benefits of “regular physical exercise, body weight management, healthy diet, adequate sleep patterns, circadian entrainment and stress management” concluded that these comprehensive lifestyle treatments lower the risk of obesity, insulin resistance and diabetes by restoring redox homeostasis and immune-system function [102]. The restored redox homeostasis also slows down the oxidative-stress-dependent shortening of telomeres (which is related to extension of the healthy lifespan [103]), and acts through epigenetic changes [104].

On the other hand, profound changes in just a single lifestyle factor are unlikely to be equally effective. Interventions that inherently combine different components are indeed emerging as particularly effective. Therapies/activities that effectively combat chronic inflammation tend to be the ones that naturally combine several lifestyle factors. An example is yoga, which combines physical activity with stress reduction (see above). Another example is horticultural therapy (gardening) that combines physical activity with stress reduction, nature exposure, and even social interaction. Such horticultural therapy has been shown to reduce markers of stress and chronic inflammation in older adults [105,106]. Moreover, even within a single subcategory such as diet, additive and synergistic effects are key, as suggested by the benefits of the Mediterranean and MIND diets (see above [63] Alzheimer’s disease), and aptly described by Liu: “The benefit of a diet rich in fruits, vegetables, and whole grains is attributed to [their] complex mixture of phytochemicals. This explains why no single antioxidant can replace the combination of natural phytochemicals” [1].

### 4.5. Additional Factors and Connections

Any additional environmental factors that trigger ROS production can contribute to chronic inflammation. Such factors include a wide variety of environmental toxins, including pollution of air, water, and food with organic toxins, heavy metals, and other agents (for recent reviews, see [107,108,109]). This connection reveals that individual lifestyle choices are not sufficient to battle chronic inflammation and the resulting elevated disease risk. Rather, system-wide societal change is needed to lower exposure to toxins and chronic stress as well as increase access to nutritious food, to the opportunity for regular physical activity, and to education about emerging knowledge on the link between lifestyle and health. This realization brings into focus socioeconomic and geographic inequalities in the access to a health-promoting lifestyle.

## 5. Conclusions and Future Directions

Essential dietary micronutrients are required to support membrane function and prevent chronic inflammation, especially in the human brain. Membrane-soluble micronutrients (zeaxanthin, lutein, vitamin E, and omega-3 PUFAs) interact synergistically among each other and with water-soluble antioxidant metabolites (vitamin C and various phenolics) and antioxidant enzymes. Several of these dietary micronutrients have multiple essential functions as antioxidants, anti-inflammatories, and/or membrane stabilizers. The synergy among different types of micronutrient antioxidants consists of intertwined oxidation-reduction cycles that protect membrane function and allow a controlled, well-orchestrated inflammation response that does not escalate to chronic inflammation. More research is needed to assess localization and interaction among different classes of micronutrients in vivo, and to test for causal relationships via clinical trials using supplementation with combinations of micronutrients. Additional lifestyle factors (physical activity, psychological stress, and toxins) interact with dietary micronutrients in affecting the internal balance (redox homeostasis) of ROS and antioxidants. More research is needed to test interventions that combine diet, physical activity, and stress reduction. Overall, multi-component interventions should be a target of future research as well as of recommendations for lifestyle changes. At this time, it appears safe to recommend a combination of a balanced diet (modeled after whole food diets like the Mediterranean and/or MIND diet) with regular moderate exercise and stress-reduction practice. On the other hand, recommendations for any specific supplements, especially in higher doses [49,66], require expert guidance and should be revised as informed by the results of additional research.

## Figures and Tables

**Figure 1 molecules-26-05385-f001:**
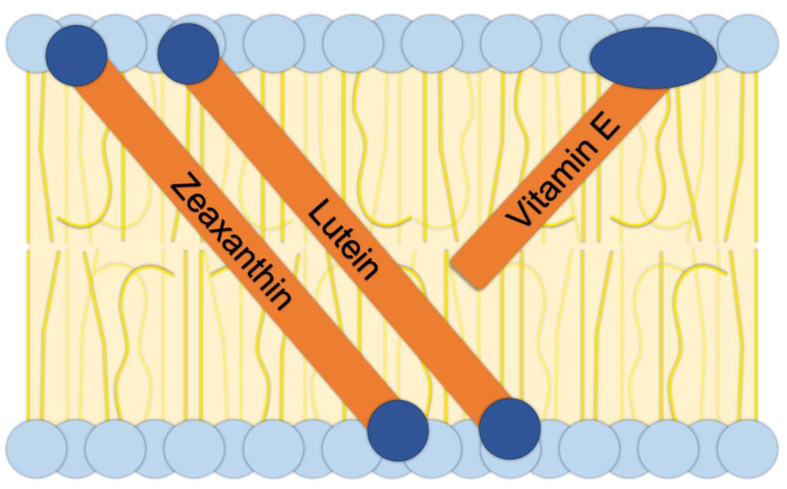
Schematic diagram depicting a biological membrane consisting of a phospholipid bilayer with embedded dietary micronutrients. The xanthophylls zeaxanthin and lutein span the entire width of the membrane, while vitamin E does not span the entire width. Hydrophobic portions of membrane phospholipids and of the embedded molecules are shown in yellow and orange, respectively. Hydrophilic membrane and micronutrient portions are shown in light blue and dark blue, respectively.

**Figure 2 molecules-26-05385-f002:**
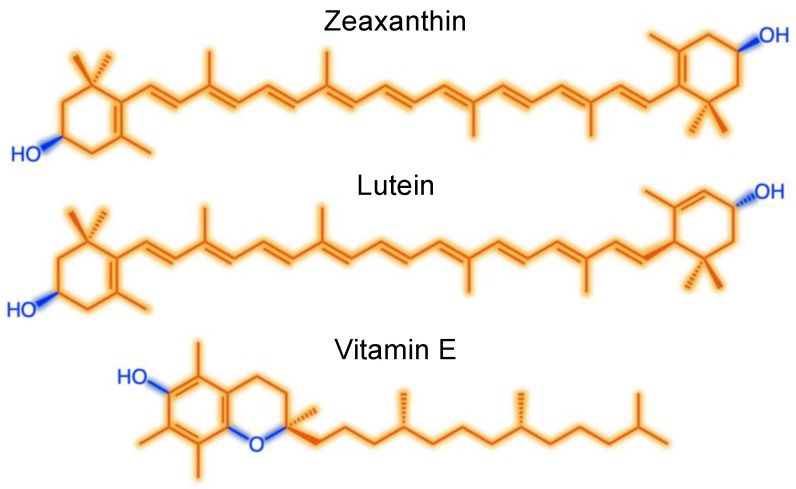
Chemical structures of the xanthophylls zeaxanthin (β,β-carotene-3,3′-diol) and lutein (β,ε-carotene-3,3′-diol) as well as vitamin E (a-tocopherol), with hydrophobic portions shown in orange and hydrophilic (oxygen-containing) portions in blue. For further detail on structure, biosynthesis, and food sources of these xanthophyll, see [6].

**Figure 3 molecules-26-05385-f003:**
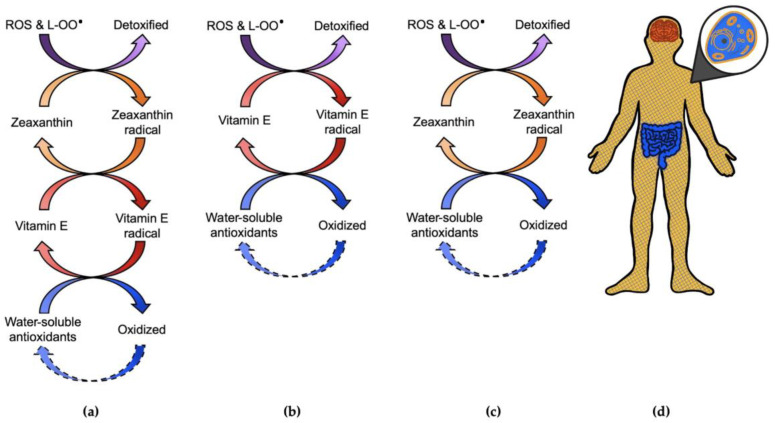
(**a**) Schematic depiction of the intertwined oxidation-reduction cycles that allow recycling (via re-reduction) of zeaxanthin radical by vitamin E after zeaxanthin has detoxified (reduced) ROS and/or lipid peroxyl radicals (L-OO^•^), followed by recycling of vitamin E radical by water soluble antioxidants. (**b**) Schematic depiction of recycling of vitamin E radical by water soluble antioxidants. (**c**) Schematic depiction of direct recycling of zeaxanthin radical by water soluble antioxidants. (**d**) Outline of the human body highlighting the gut with water-soluble dietary polyphenols (dark blue), cells (top-right magnification) throughout the body with aqueous cytosol and inner and outer membranes (light orange with blue hatches), and a high concentration of membranes with their lipid-soluble antioxidants in the brain (dark orange).

**Figure 4 molecules-26-05385-f004:**
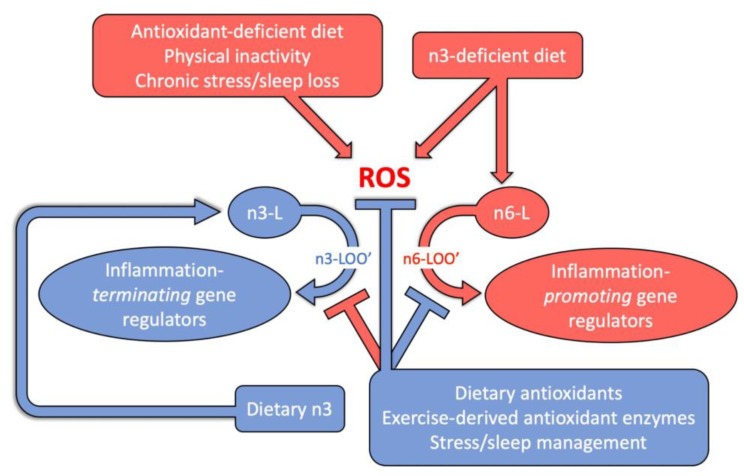
Schematic depiction of interactions between diet and other lifestyle factors in modulating reactive oxygen species (ROS) and regulators of the immune response derived from oxidation products of membrane phospholipids containing either omega-3 fatty acids (n3-L) or omega-6 fatty acids (n6-L). Red boxes and arrows designate inflammation-promoting factors and responses, while blue boxes and arrows designate inflammation-terminating factors or responses. Note that dietary antioxidants have the dual effect of dampening the production of inflammation-promoting n6-L-based regulators (blue T-shaped line) as well as dampening the production of inflammation-terminating n3-L-based regulators (red T-shaped line).

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
