# Peer review of "Synergistic Action of Membrane-Bound and Water-Soluble Antioxidants in Neuroprotection"

_molecules, 2021, doi:10.3390/molecules26175385_

Round 1

Reviewer 1 Report

In this paper, Polutchko and colleagues investigated the role of different classes of dietary micronutrients in protecting membrane function to prevent chronic inflammation. However, most of the review focuses only on clinical trial, while experimental evidence (both in vitro and in vivo) regarding the neuroprotective effects of both membrane-bound and water-soluble anti-oxidants are lacking. Within this frame, it would be interesting to know if the Authors have considered to extend the discussion (especially in Paragraph 4.1.) regarding the neuroprotective effects, the mechanism of action and the interaction among different classes of micronutrients in vivo in different animal/experimental models, as well as experimental evidence regarding the combination of diet, physical activity and stress reduction in maintaining oxidant-antioxidant balance. This would add significance to the paper.

There are some minor issues, which are reported below, that need to be addressed or fixed.

Abstract:

- Line 20: “MIND diet”; please spell out the abbreviation;

- Lines 21-23: please rephrase.

- Line 22 “in-vivo”, line 75 “in vivo”;

Introduction:

- Figure 1: please correct the cartoon; maybe there was a problem when adding the figure within the text since it contains several big white squares hiding parts of the figure and text; moreover, the Authors should better illustrate the oxygen-containing portions of zeaxanthin and lutein, which are of critical importance for the function of these two xanthophylls.

- Lines 118-122; please rephrase.

The English language and style are fine, although some parts can be rephrased since they contain very long and articulated sentences. In any case, a minor spell check is required throughout the manuscript.

Author Response

Reviewer comment: it would be interesting to know if the Authors have considered to extend the discussion (especially in Paragraph 4.1.) regarding the neuroprotective effects, the mechanism of action and the interaction among different classes of micronutrients in vivo in different animal/experimental models, as well as experimental evidence regarding the combination of diet, physical activity and stress reduction in maintaining oxidant-antioxidant balance. This would add significance to the paper.

RESPONSE: WE HAVE EXTENDED THE DISCUSSION IN PARAGRAPH 4.1 BY ADDING THE FOLLOWING TEXT:

“An active area of research on the benefits of supplementation, with a focus on zeaxanthin, is the prevention of eye disease, such as age-related macular degeneration. Combinations of zeaxanthin and/or lutein with synergistically acting lipid-soluble vitamin E, water-soluble vitamin C, and/or omega-3 polyunsaturated fatty acids (e.g., DHA) were particularly effective (Marshall and Roach 2013; Zampatti et al. 2014; McCusker et al. 2016). The mechanisms of action of zeaxanthin and lutein in this context have been documented and reviewed extensively (Bone et al 1988, 1993; Thomson et al. 2002; Toyoda et al. 2002; Niedzwiedzki et al. 2010; see also Demmig-Adams and Adams 2013; Demmig-Adams et al. 2020).”

as well as (for the case of the Mediterranean diet), “The mechanisms underlying this effect include amelioration of chronic inflammation as evidenced by lower levels of inflammatory markers (Calder et al. 2011; Fontana and Partridge 2015).”

and (for the synergy among different lifestyle factors), “The benefits on brain function provided by a combination of several lifestyle factors, including physical activity, cognitive engagement, and diet, have been reviewed previously (Phillis 2017). Another recent review of the synergistic benefits of “regular physical exercise, body weight management, healthy diet, adequate sleep patterns, circadian entrainment and stress management” concluded that these comprehensive lifestyle treatments lower the risk of obesity, insulin resistance and diabetes by restoring redox homeostasis and immune-system function (Valenzuela et al. 2021). The restored redox homeostasis also slows down the oxidative-stress-dependent shortening of telomeres (which is related to extension of the healthy lifespan; Qiao et al. 2020) and acts through epigenetic changes (Kaliman et al. 2019).

Reviewer comment: There are some minor issues, which are reported below, that need to be addressed or fixed. Abstract: - Line 20: “MIND diet”; please spell out the abbreviation

RESPONSE: THIS HAS BEEN DONE

Reviewer comment - Lines 21-23: please rephrase.

RESPONSE: THIS HAS BEEN REPHRASED AND ALSO CUT AS REQUESTED BY REVIEWER 2. THE TEXT NOW READS: “Effective whole-food-based diets include the Mediterranean and the MIND diet (Mediterranean-DASH Intervention for Neurodegenerative Delay diet, where DASH stands for Dietary Approaches to Stop Hypertension).”

Reviewer comment: - Line 22 “in-vivo”, line 75 “in vivo”;

RESPONSE: THIS IS PART OF THE SENTENCE THAT WAS CUT AS REQUESTED BY REVIEWER 2.

Reviewer comment: Introduction: - Figure 1: please correct the cartoon; maybe there was a problem when adding the figure within the text since it contains several big white squares hiding parts of the figure and text; moreover, the Authors should better illustrate the oxygen-containing portions of zeaxanthin and lutein, which are of critical importance for the function of these two xanthophylls.

RESPONSE: WE HAVE REVISED FIGURE 1 TO ELIMINATE THIS PROBLEM AND BETTER HIGHLIGHT THE OXYGEN-CONTAINING PORTIONS OF ZEAXANTHIN, LUTEIN, AND VITAMIN E.

Reviewer comment: - Lines 118-122; please rephrase.

RESPONSE: THIS SECTION HAS BEEN SPLIT INTO TWO SENTENCES AND REVISED, AND NOW READS: “In fact, it is critical that vitamin E and zeaxanthin radicals be recycled (by re-reduction) because they themselves are potentially harmful oxidants [18,19]. The importance of this recycling process is illustrated by the fact that the antioxidant capacity of various vitamin E constituents is determined by how efficiently they can be recycled [19].”

Reviewer comment: The English language and style are fine, although some parts can be rephrased since they contain very long and articulated sentences. In any case, a minor spell check is required throughout the manuscript.

RESPONSE WE HAVE SIMPLIFIED AND BROKEN UP A NUMBER OF LONG SENTENCES, AND MADE MINOR EDITS THROUGHOUT THE TEXT.

Reviewer 2 Report

This paper by Polutchko et al reviewed the synergistic effects of different classes of dietary micronutrients in supporting membrane structure and function. Especially, authors discussed that essential dietary micronutrients are required to support the function of biological membranes and prevent chronic inflammation in the human brain. The review allowed me to understand the role of dietary micronutrients. The general purpose of this study is clear. Overall I recommend this paper to be published in Molecules after addressing the following comments.

Major point
1. Abstract: I feel that it is desirable not to be described future prospects in the abstract section. I would ask that you reconsider this point.

2. Figure 1 Please confirm that Figure one more time. It looks like the background colors are overlapping.

Minor point

1. Abbreviations: Abbreviations used should be defined once the first time they appear in the text.

   L20 You should spell out or define MIND.

  L109 oxidants (ROS) → oxidants (such as ROS) directly~

  L130 You should spell out or define NADH

  L184 ~detoxified (reduced) reactive oxygen species (ROS)~ → ~detoxified (reduced) ROS~

2. Font: L247 ~L277

  You should edit the document to make it all the same font.

Author Response

Reviewer comment: This paper by Polutchko et al reviewed the synergistic effects of different classes of dietary micronutrients in supporting membrane structure and function. Especially, authors discussed that essential dietary micronutrients are required to support the function of biological membranes and prevent chronic inflammation in the human brain. The review allowed me to understand the role of dietary micronutrients. The general purpose of this study is clear. Overall I recommend this paper to be published in Molecules after addressing the following comments. Major point
1. Abstract: I feel that it is desirable not to be described future prospects in the abstract section. I would ask that you reconsider this point.

RESPONSE: THIS STATEMENT HAS BEEN DELETED.

Reviewer comment: Figure 1: Please confirm that Figure one more time. It looks like the background colors are overlapping.

RESPONSE THIS FIGURE HAS BEEN REVISED.

Reviewer comment: Minor point. 1. Abbreviations: Abbreviations used should be defined once the first time they appear in the text.

RESPONSE: THIS CHANGE HAS BEEN MADE.

Reviewer comment:L20 You should spell out or define MIND.

RESPONSE: THIS HAS BEEN DONE.

Reviewer comment: L109 oxidants (ROS) → oxidants (such as ROS) directly~

RESPONSE: THIS CHANGE HAS BEEN MADE.

Reviewer comment: L130 You should spell out or define NADH

RESPONSE: THIS CHANGE HAS BEEN MADE.

Reviewer comment: L184 ~detoxified (reduced) reactive oxygen species (ROS)~ → ~detoxified (reduced) ROS~

 RESPONSE: THIS CHANGE HAS BEEN MADE.

Reviewer comment: Font: L247 ~L277, You should edit the document to make it all the same font.

RESPONSE: THIS CHANGE HAS BEEN MADE, AND THE WHOLE DOCUMENT EDITED TO HAVE THE SAME FONT.